# Hyperkalemic or Low Potassium Cardioplegia Protects against Reduction of Energy Metabolism by Oxidative Stress

**DOI:** 10.3390/antiox12020452

**Published:** 2023-02-10

**Authors:** Hongting Diao, Haiwei Gu, Qin M. Chen

**Affiliations:** 1Department of Pharmacy Practice and Science, College of Pharmacy, The University of Arizona, Tucson, AZ 85721, USA; 2College of Health Solutions, Arizona State University Phoenix, Phoenix, AZ 85004, USA; 3Center for Translational Science, Florida International University, 11350 SW Village Parkway, Port St. Lucie, FL 34987, USA

**Keywords:** cardiopulmonary bypass, cardioplegic solution, potassium, reactive oxygen species, energy metabolism

## Abstract

Open-heart surgery is often an unavoidable option for the treatment of cardiovascular disease and prevention of cardiomyopathy. Cardiopulmonary bypass surgery requires manipulating cardiac contractile function via the perfusion of a cardioplegic solution. Procedure-associated ischemia and reperfusion (I/R) injury, a major source of oxidative stress, affects postoperative cardiac performance and long-term outcomes. Using large-scale liquid chromatography–tandem mass spectrometry (LC-MS/MS)-based metabolomics, we addressed whether cardioplegic solutions affect the baseline cellular metabolism and prevent metabolic reprogramming by oxidative stress. AC16 cardiomyocytes in culture were treated with commonly used cardioplegic solutions, High K^+^ (HK), Low K^+^ (LK), Del Nido (DN), histidine–tryptophan–ketoglutarate (HTK), or Celsior (CS). The overall metabolic profile shown by the principal component analysis (PCA) and heatmap revealed that HK or LK had a minimal impact on the baseline 78 metabolites, whereas HTK or CS significantly repressed the levels of multiple amino acids and sugars. H_2_O_2_-induced sublethal mild oxidative stress causes decreases in NAD, nicotinamide, or acetylcarnitine, but increases in glucose derivatives, including glucose 6-P, glucose 1-P, fructose, mannose, and mannose 6-P. Additional increases include metabolites of the pentose phosphate pathway, D-ribose-5-P, L-arabitol, adonitol, and xylitol. Pretreatment with HK or LK cardioplegic solution prevented most metabolic changes and increases of reactive oxygen species (ROS) elicited by H_2_O_2_. Our data indicate that HK and LK cardioplegic solutions preserve baseline metabolism and protect against metabolic reprogramming by oxidative stress.

## 1. Introduction

Cardiovascular disease is the principal cause of mortality worldwide. Most cardiac patients suffering from coronary artery disease have plaque buildup in the wall of the coronary arteries. With advancements in medical technology, an occluded coronary artery can be re-opened by the percutaneous coronary intervention technique. Open-heart surgery remains an option of treatment, especially for multivessel disease, diffuse or complex coronary disease, with or without valvular disease [1,2,3]. The development of cardiopulmonary bypass (CPB) using a heart–lung machine to sustain systematic circulation during operation has revolutionized cardiac surgery and reduced the mortality of patients with coronary artery disease. However, approximately 67% of CPB patients have post-operative complications, including atrial fibrillation, pulmonary dysfunction, or renal failure [4]. One contributing factor to these types of organ injury is procedure-associated ischemia and reperfusion (I/R), which produce oxidative stress.

In order to achieve surgical precision, cardiac contraction is minimized by the perfusion of a cardioplegic solution. This procedure causes controlled cessation of blood flow to the myocardium, i.e., myocardial ischemia. Upon completion of the surgical procedure, cardiac contraction and blood flow are resumed. Either ischemia or reperfusion results in an increased production of reactive oxygen species (ROS) [5,6]. Ischemia results in incomplete electron transport, impairment of mitochondrial respiration, and mitochondrial ROS production [7,8,9]. Reperfusion activates xanthine oxidase and NADPH oxidase in cardiac cells, which already inherit impaired mitochondria from ischemia, causing further increases of ROS [6,10]. Elevated levels of lipid peroxidation products, such as malondialdehyde, have been detected in the myocardium of patients following I/R from the coronary artery bypass graft surgery (CABG) [11]. Preventing oxidative stress is expected to reduce surgical-procedure-associated complications.

Cardioplegic solutions, along with induced hypothermia, during open-heart surgery are designed to prevent many detrimental effects of I/R [12,13,14]. The most commonly used cardioplegic solutions include modified St. Thomas, i.e., high potassium (high K^+^, HK) and low potassium (low K^+^, LK), Del Nido (DN), and histidine–tryptophane–ketoglutarate Custodiol^®^ (HTK). Celsior (CS) was developed as a cardioplegia as well as a solution for donor organ preservation during cardiac transplant [15]. The HK, DN, and CS cardioplegic solutions, also known as extracellular cardioplegia, are hyperkalemic, delivering K^+^ via the coronary vasculature to reach the localized concentration of 15 to 25 mM [16]. Such high concentrations of K^+^ cause the resting membrane potential of the myocardium to change from −80 mV to −50 mV [17]. When the membrane potential reaches −50 mV, the voltage-dependent Na^+^ channels are inactivated, preventing the Na^+^-induced spike and rapid propagation of the action potential, thereby causing depolarized diastolic cardiac arrest. LK is used following the administration of HK to maintain the diastolic arrest. HTK, also called intracellular cardioplegia, is low in Na^+^. Depletion of Na^+^ in the extracellular spaces produces a hyperpolarization of the plasma membrane of cardiomyocytes, causing cardiac arrest in diastole [18]. Although these cardioplegic solutions have been widely used for open-heart surgery worldwide, little is known about the impact of these agents on the metabolism of cardiomyocytes.

Metabolomics provides an effective method for measuring overall chemical intermediates or metabolic state [19,20]. The application of metabolomics will not only help to identify new biomarkers for metabolic complications associated with cardioplegia but also will allow us to provide scientific evidence for choosing an ideal cardioplegic solution for cardiothoracic surgery, as such evidence is lacking. In this study, using large-scale liquid chromatography–tandem mass spectrometry (LC-MS/MS)-based metabolomics and AC16 human cardiomyocytes, we evaluated the metabolic impact of five commonly used cardioplegic solutions and addressed whether any of these solutions prevent metabolic changes by oxidative stress.

## 2. Material and Methods

### 2.1. Reagents

Cardioplegic solutions of HK, LK, DN, HTK (HTK Custodiol^®^, Bretschneider, Cardiolink Group, Barcelona, Spain), and CS (Sanofi, Bridgewater, NJ, USA) were collected as leftovers from cardiothoracic surgeries at the Banner-University Medical Center Tucson or Phoenix Children’s Hospital. HK, LK, and DN were formed at Banner-University of Arizona Hospital Pharmacy from Plasma-Lyte A (Baxter Healthcare Corporation, Deerfield, IL, USA).

### 2.2. Cell Lines, Culture Conditions, and Treatments

AC16 human cardiomyocytes were purchased from Millipore Sigma-Aldrich and cultured in Dulbecco’s modified Eagle’s medium mixed with Ham’s F12 (DMEM/F12, GIBCO, Life Technologies, Carlsbad, CA, USA) supplemented with 12.5% fetal bovine serum (FBS, R&D Systems, Minneapolis, MN, USA) and antibiotics (1% of 100 U penicillin/50 g/mL streptomycin, GIBCO) for culture in a 5% CO_2_ incubator at 37 °C with weekly subculture. The cells were seeded at a density of 0.3 × 10^6^ cells per well in 6-well plates and grown to 90% confluence before experiments.

Cells were treated with cardioplegic solutions according to the ratios used clinically. HK or LK (0.4 mL) was mixed with 1.6 mL DMEM, whereas DN (1.6 mL) was mixed with 0.4 mL DMEM. For HTK or CS, 2 mL pure crystalloid solution was added to AC16 cells. Cells in 6-well plates were serum-starved overnight in 0.5%FBS/DMEM/F12 and treated with a cardioplegic solution for 3 h followed by exposure to 200 μM H_2_O_2_ for 1 h.

### 2.3. Metabolomics

The samples were collected for LC-MS/MS metabolomics as described [21,22,23,24]. Following a quick rinse with phosphate-buffered saline (PBS), cells were placed in 1 mL/well of 8:2 (*v*:*v*) methanol: H_2_O for 30 min incubation on dry ice. This serves to quench metabolism and extract the metabolites. The cells were scraped from the culture plates and transferred to centrifuge tubes. Another 0.7 mL/well of 8:2 (*v*:*v*) methanol: H_2_O was added to the plates on the dry ice to scrape and combined with the corresponding centrifuge tubes. Cell-free extracts were collected after 10 min centrifugation at 13,000 rpm under 4 °C. The soluble fractions of the extracts were dried at 4 °C using a speed vacuum. The samples were processed for LC-MS/MS as described [25].

LC-MS/MS analyses were performed using an Agilent 1290 UPLC-6490 QQQ-MS system (Santa Clara, CA, USA). Each sample was injected twice by an auto-sampler set at 4 °C, 10 μL for analysis using a negative ionization mode or 4 μL for analysis using a positive ionization mode. Both chromatographic separations were performed in the hydrophilic interaction chromatography mode on a Waters XBridge BEH Amide column (150 × 2.1 mm, 2.5 μm particle size, Waters Corporation, Milford, MA, USA), with a flow rate of 0.3 mL/min and the column compartment temperature of 40 °C. The mobile phase contained Solvent A (10 mM ammonium acetate and 10 mM ammonium hydroxide in 95% H_2_O/5% ACN) and Solvent B (10 mM ammonium acetate and 10 mM ammonium hydroxide in 5% H_2_O/95% ACN). After an initial 1 min isocratic elution of 90% Solvent B, the percentage of Solvent B decreased to 40% by *t* = 11 min. Solvent B was maintained at 40% for 4 min (*t* = 15 min) before gradually returning to 90% in preparation for the next injection. The mass spectrometer was equipped with an electrospray ionization source, and targeted data acquisition was performed in the multiple-reaction monitoring (MRM) mode. Agilent Masshunter Workstation software (Santa Clara, CA, USA) was used to control the operation of the LC-MS/MS system, whereas the extracted MRM peaks were integrated using the Agilent MassHunter Quantitative Data Analysis (Santa Clara, CA, USA).

### 2.4. Cell Morphology

AC16 cells were seeded in 6-well plates for treatments as described in Section 2.2. The cells were fixed in 4% paraformaldehyde for 5 min, washed twice with PBS, and stained with 0.1% Coomassie blue (dissolved in 10% acetic acid and 50% methanol) for 10 min. After washing off the dye, cell morphology was recorded under an inverted microscope with 10× lens (Rebel, Echo, San Diego, CA, USA).

### 2.5. ROS Detection

AC16 cells were seeded in a clear bottom 96-well dark plate for treatment with cardioplegic solutions or various concentrations of lidocaine for 3 h. After rinsing two times with warm (37 °C) Hanks’ Balanced Salt Solution (HBSS, 1.26 mM CaCl_2_, 0.49 mM MgCl_2_, 0.406 mM MgSO_4_, 5.33 mM KCl, 0.44 mM KH_2_PO_4_, 4.17 mM NaHCO_3_, 137.93 mM NaCl, and 0.338 mM Na_2_HPO_4_), the cells were incubated in dark with 10 µM 2′,7′–dichlorofluorescein diacetate (DCFDA) for 30 min at 37 °C in a 5% CO_2_ incubator. After the removal of DCFDA, 100 µL HBSS was added to the cells along with 200 µM H_2_O_2_ for measurements of ROS according to the manufacturer’s instructions (ROS detection kit, Abcam, Cambridge, UK). DCF fluorescence was recorded at Ex485/Em535 nm by a plate reader (CLARIOstar Plus, BMG Labtech, Ortenberg, Germany).

### 2.6. Statistical Analysis

Changes in metabolites or pathways between control versus treatment were analyzed using the web-based software MetaboAnalyst 5.0 (https://www.metaboanalyst.ca/MetaboAnalyst/ModuleView.xhtml, accessed 3 June 2022). PCA, K-means, heatmap clustering, and pathway analysis overview of altered metabolic profiles were analyzed using MetaboAnalyst 5.0. Bar graphs were presented as means ± standard deviations (SD) of the relative abundance of metabolites and analyzed via one-way ANOVA and corrected by Dunnett’s multiple comparisons test with GraphPad Prism 9 software. The data with two grouping variables were analyzed with two-way ANOVA and corrected by Turkey’s multiple comparisons test. *p* value or adjusted *p* value < 0.05 was set as the threshold for significant difference.

## 3. Results

### 3.1. Effects of Cardioplegic Solutions on the Metabolome

Table 1 lists the components of five cardioplegic solutions commonly used in the US and internationally for CPB surgeries (HK, LK, DN, and HTK) or cardiac transplants (CS). To evaluate the impact of these solutions on cellular metabolism, we treated AC16 cardiomyocytes according to their clinical applications, with the final concentration of each component listed in Table 1. The length of treatment was 4 h, mimicking the average CPB surgical time. A total of 94 metabolites were detected among the control (Ctrl) and cells treated with a cardioplegic solution, as shown in the heatmap (Figure 1A). When compared with the Ctrl, a similar pattern of metabolites was found in HK- and LK-treated groups, but HTK- and CS-treated cells exhibited a clear difference, showing decreases in 12 amino acids and 4 sugars (Figure 1A). The metabolic profile of DN-treated cells is closer to that of Ctrl, HK, or LK (Figure 1A,B). The overall metabolic profiles as summed by the principal component analysis (PCA) revealed that HK- or LK-cardioplegia-treated cells showed no significant difference from the Ctrl (Figure 1B).

PCA indicates significant deviations in the profile of HTK- or CS-treated cells (Figure 1B). The differences in the metabolites are displayed in a volcano plot (Figure 2A,B). The pathway analyses showed that HTK and CS caused significant perturbations of the pathways of: (1) alanine, aspartate, and glutamate metabolism; (2) histidine metabolism; (3) arginine biosynthesis; (4) tricarboxylic acid (TCA) cycle; (5) aminoacyl-tRNA biosynthesis; and (6) valine, leucine, and isoleucine biosynthesis (Figure 2C,D). CS also affected the glyoxylate and dicarboxylate metabolism pathway (Figure 2D). These results indicate that unlike HTK or CS, HK or LK cardioplegic solution preserves the basal metabolic profile.

### 3.2. Metabolic Profiles of Oxidative Stress and Impact of Cardioplegic Solutions

H_2_O_2_ at the dose of 200 μM was chosen here for metabolomics experiments since it does not cause detectable cytotoxicity, as measured by cell morphology, cell number, or general metabolic activity via CCK-8 assay [24,26]. This dose mimics the range of oxidants in the myocardium during I/R, 10–30 μM, based on the common belief that intracellular oxidants are 10–100-fold lower than extracellular H_2_O_2_ concentration [27,28].

A significant separation of metabolic profiles was discovered between the Ctrl and H_2_O_2_-treated group by PCA (Figure 3A). The heatmap shows 35 metabolites that were significantly up- (16) or down- (19) regulated, as determined by the Student’s *t*-test with FDR correction (Figure 3B). Among the significantly decreased metabolites are NAD and acetylcarnitine (Figure 3B). Increased metabolites include glucose derivatives, i.e., glucose 6-P, glucose 1-P, fructose, mannose, and mannose 6-P. Several sugar alcohols in the pentose phosphate pathway also increased, including L-arabitol, adonitol, Ribose 5-P, and xylitol (Figure 3B). Pathway analyses revealed significant alteration of nine metabolic pathways by H_2_O_2_ treatment (*p* < 0.05): (1) nicotinate and nicotinamide metabolism; (2) pentose phosphate pathway; (3) TCA cycle; (4) amino sugar and nucleotide sugar metabolism; (5) pentose and glucuronate interconversions; (6) alanine, aspartate, and glutamate metabolism; (7) purine metabolism; (8) glycolysis; and (9) butanate metabolism (Figure 3C).

Compared with the Ctrl or H_2_O_2_-treated cells, pretreatment with HK or LK was able to shift the metabolic profile of H_2_O_2_-treated cells towards that of Ctrl (Figure 4A). Among the 17 metabolites of 9 significantly changed pathways due to H_2_O_2_ treatment, HK or LK pretreatment was able to block H_2_O_2_ from inducing increases or decreases in the majority of these metabolites (Figure 4B). Specifically, the decrease in NAD or acetylcarnitine was attenuated by HK or LK treatment (Figure 5A,B). H_2_O_2_-induced increases in glucose 6-P, glucose 1-P, mannose 6-P, and oxaloacetic acid were inhibited by HK or LK (Figure 5B,C). The increases in the metabolites in the pentose phosphate pathway, i.e., xylitol, L-arabitol, adonitol, and D-ribose 5-P, due to H_2_O_2_ treatment were prevented by HK or LK (Figure 5D). These data indicate the capacity of HK or LK for protection against metabolic shifts by oxidative stress.

### 3.3. Protection against ROS Generation

To explore the mechanism of HK or LK for protection against metabolic shift by oxidative stress, we tested whether the cardioplegic solutions affect ROS production by H_2_O_2_. AC16 cells were able to tolerate five cardioplegic solutions without morphological evidence of toxicity (Figure 6A). When the baseline ^•^OH generation was measured by treatment of a cardioplegic solution using DCFH-DA assay, the level of DCF fluorescence in HK- or LK-treated cells was similar to that of the control without any treatment (Figure 6B). The positive control of 200 μM of H_2_O_2_ showed a time-dependent elevation of DCF fluorescence over 240 min (Figure 6B). DN, HTK, or CS induced elevation of DCF fluorescence, with CS producing nearly half the amount of ROS as H_2_O_2_ (Figure 6B). We then investigated whether any of these cardioplegic solutions are capable of preventing H_2_O_2_ from generating ^•^OH. When pretreating cells with a cardioplegic solution, we found an inhibition of H_2_O_2_ induced ^•^OH generation by HK or LK, but not DN, HTK, or CS (Figure 6C).

## 4. Discussion

Using LC-MS/MS-based metabolomics, we evaluated the impact of five cardioplegic solutions on cellular metabolism and metabolic alterations by oxidative stress. Our data indicate that HK and LK cardioplegia solutions did not cause significant changes in the baseline metabolic profile, whereas HTK and CS induced significant metabolic shifts. Induction of oxidative stress by H_2_O_2_ treatment resulted in decreases in NAD and acetylcarnitine but increases in glucose 6-P, glucose 1-P, mannose 6-P, oxaloacetic acid, and four sugar alcohols from the pentose phosphate pathway. Our data suggest that HK and LK cardioplegic solutions can provide protection against alterations in energy metabolism due to oxidative stress at the cellular level. Such protection correlates with an inhibition of ROS generation.

The cardioplegic solutions are capable of supporting an extended period of myocardial ischemia with organ preservation during cardiac surgery. The choice of which cardioplegic solution is based on the preference and experience of each surgeon. HK, LK, and DN, the so-called blood cardioplegia, require isogenic mixing of the patient’s blood for their clinical application, whereas HTK or CS is perfused into the myocardium as a pure crystalloid. Experimental or clinical evidence suggests that blood cardioplegia is superior to the crystalloids for cardiac protection [29,30,31,32,33,34,35]. Among the many advantages of blood cardioplegia are oxygen-carrying capacity, continuous supply of metabolic substrates, physiologic buffering, and consistent osmotic pressure [36,37]. We mimicked the clinical application by mixing HK, LK, and DN with the culture medium. This approach has maintained the supply of nutrients, including glucose, amino acids, vitamins, and minerals, which were absent in the crystalloid solution HTK or CS. Therefore, the large deviation in the basal metabolic profile by HTK or CS could result from nutrient deprivation in the cell culture experiments. Nevertheless, our finding of HK or LK having a minimal impact on the baseline metabolic profile yet protecting against metabolic switch by oxidants supports the clinical observation of the superiority of the blood cardioplegia.

The metabolic shift detected here by oxidative stress resembles that of surgery-induced ischemia. A decrease in NAD level has been reported in myocardial tissue due to CABG surgery [38,39]. Metabolic reprogramming indeed occurs with CPB since a nuclear magnetic resonance study of the left atrial tissue showed increases in glucose, pyruvate, citrate, and lactate, indicating a reduction of glycolytic energy metabolism [40]. On the other hand, increases in sugar alcohols have been linked to apoptosis of cardiomyocytes during ischemia [41,42,43]. Although xylitol, adonitol, arabitol, and ribose-5-P may not be inducers of apoptosis, the correlation of their elevations with cell death suggests the functionality as a biomarker of cell injury. Inhibition of H_2_O_2_-induced metabolic changes by HK and LK suggest the cytoprotective capacity of HK and LK against ischemic injury or oxidative stress.

Ischemia causes a shift from aerobic to anaerobic metabolism, where ATP production is decreased due to O_2_ deprivation for the respiratory chain. Glycolysis, the main ATP source in the ischemic myocardium, is progressively limited by the restraint of glyceraldehyde 3-phosphate dehydrogenase flux due to NADH accumulation [44]. High levels of NADH and FADH_2_ inhibit β-oxidation, leading to an accumulation of long-chain acyl CoA, acylcarnitine esters, and free fatty acids [45,46]. These lipophilic compounds can act as “detergents”, altering membrane functions, such as Na^+^/K^+^ ATPase, and sarcoplasmic reticular Ca^2+^-stimulated ATPase [47]. Upon reperfusion, these changes contribute to the elevation of cytosolic Ca^2+^ [48]. Oxidative stress can cause Ca^2+^ influx and elevation of cytosolic Ca^2+^. Elevation of cytosolic Ca^2+^ either by ischemic reperfusion or oxidative stress causes mitochondrial membrane permeability transition and a further increase in oxidant production [5,6].

HK and LK deliver 20 mg/L or 0.085 mM lidocaine, whereas DN introduces 104 mg/L or 0.444 mM lidocaine to the cells in the culture. In contrast, lidocaine is absent in HTK and CS. Lidocaine is a sodium channel blocker that inhibits the entry of Na^+^ into cells through a fast voltage-gated Na^+^ channel (Na_V_). A number of publications have reported the cytotoxicity of lidocaine in a variety of cell types in the mM concentration range [49,50]. Cytotoxicity may result from high doses that allow lidocaine to act as a surfactant and disrupt the plasma membrane [50]. This may explain the lack of full protection of DN against the metabolic change of oxidative stress.

In cardiomyocytes, lidocaine appears to serve as a cytoprotective agent. Lidocaine protects H9C2 cardiomyocytes against loss of viability by hypoxia [51]. Protection against myocardial infarction has been reported in mice with lidocaine injection [52]. A similar discovery was reported with regional ischemia in rats [53]. CPB in a canine model showed that lidocaine protected against mortality due to global myocardial ischemia and reperfusion [54]. An early study using a canine model of regional ischemic reperfusion showed a reduction of infarct size by lidocaine in association with decreased lipid peroxidation [55]. The protective effect of lidocaine against myocardial infarction was also observed in a porcine experimental model [56]. These protective effects may result from the inhibition of Na_v_ and the prevention of intracellular Na^+^ increase. Na^+^ efflux is coupled to Ca^2+^ influx via the sodium–calcium exchanger (NCX). Inhibition of intracellular Ca^2+^ elevation by lidocaine has been demonstrated by pacing isolated rat hearts [57]. In cardiomyocytes, late openings of the sodium channel by H_2_O_2_ treatment may contribute to an increase in cytosolic Ca^2+^ [58]. Lidocaine may block H_2_O_2_ from inducing elevation of cytosolic Ca^2+^ and, therefore, production of ROS from the mitochondria. A preliminary experiment indeed showed protection against ROS generation by lidocaine (Figure 7). Early evidence indicates that lidocaine preserved mitochondrial oxidative phosphorylation during ischemic reperfusion [59,60]. Consistent with the protective role of lidocaine, an inhibitor of NCX was capable of preventing intracellular Ca^2+^ overload or mitochondrial Ca^2+^ increase and enhancing cardiac recovery following ischemia [61,62]. Therefore, the observed protective effect of HK or LK may be related to the presence of lidocaine and the inhibition of ROS generation.

## Figures and Tables

**Figure 1 antioxidants-12-00452-f001:**
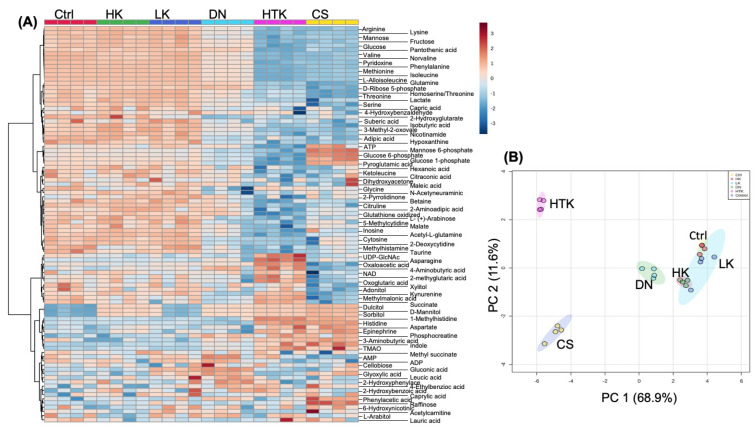
Effects of 5 Cardioplegic Solutions on Cellular Metabolism. AC16 human cardiac myocytes were seeded in 6-well plates and cultured in DMEM containing 12.5% FBS. When reaching 90% confluency, cells were placed in 0.5% FBS/DMEM/F12 for overnight serum starvation. The cells were treated with HK, LK, DN, HTK, and CS for 4 h before extraction of aqueous metabolites from cell lysates for LC-MS/MS-based metabolomics. The overall metabolic profile differences between control (Ctrl) and cells treated with a cardioplegia solution (n = 4) are shown in the heatmap (**A**) and by a principal component analysis (PCA) score plot (**B**).

**Figure 2 antioxidants-12-00452-f002:**
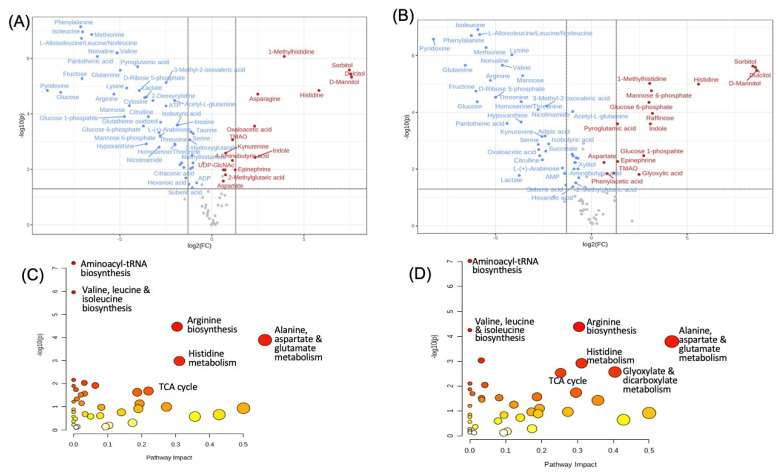
Metabolites and Metabolic Pathways Altered by HTK and CS. AC16 cells seeded in 6-well plates were used for the collection of aqueous metabolites as described in Figure 1 for LC-MS/MS-based metabolomics. A total of 67 or 65 metabolites showed statistically significant differences as determined by Student’s *t*-test (n = 4, *p* < 0.05) between control versus HTK or CS and are displayed in the volcano plots (**A**,**B**) generated by the MetaboAnalyst 5.0 software. The altered metabolic pathways are displayed with the *x*-axis representing the pathway impact score computed from pathway topological analysis and the *y*-axis being the -log10 of the *p*-value obtained from pathway enrichment analysis (**C**,**D**). The metabolic pathways perturbed by the HTK and CS treatment are displayed in the gradient from yellow to red, reflecting the *p*-value from statistical analysis, as shown in the *y*-axis, from non-significant (≤1 for −log10, *p* ≥ 0.1, light yellow) to significant (≥1 but ≤2 for −log10, *p* ≤ 0.01 to *p* ≥ 0.001, orange) to highly significant (≥2 for −log10, *p* ≤ 0.0001, red). The size of the circles reflects the number of metabolites altered in the metabolic pathway (**C**,**D**).

**Figure 3 antioxidants-12-00452-f003:**
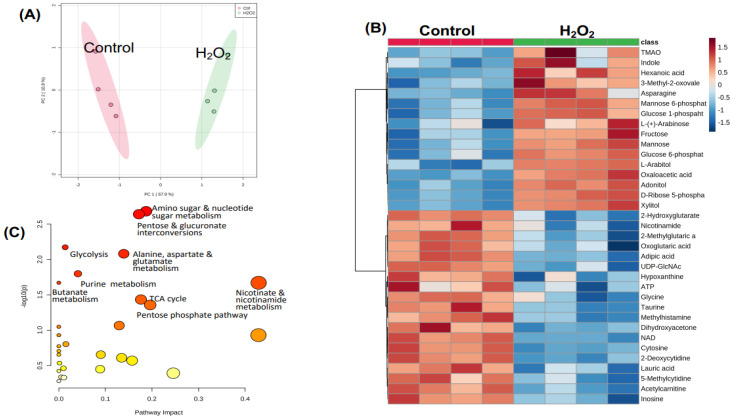
Metabolites and Metabolic Pathways Altered by H_2_O_2_ treatment. AC16 human cardiac myocytes were seeded in 6-well plates and cultured in DMEM containing 12.5% FBS. When reaching 90% confluency, cells were serum-starved in 0.5% FBS DMEM/F12 overnight. The cells were treated with 200 μM of H_2_O_2_ for 1 h before extraction of aqueous metabolites by 80% methanol from cell lysates for LC-MS/MS-based metabolomics. The overall metabolic profile differences between ctrl and H_2_O_2_-treated cells are shown by the PCA score plot (**A**). The heatmap shows significant differences (n = 4, *p* < 0.05) in 35 metabolites between Ctrl and H_2_O_2_-treated cells (**B**). The altered metabolic pathways by H_2_O_2_ treatment are displayed with the *x*-axis representing the pathway impact score computed from pathway topological analysis and the *y*-axis being -log10 of the *p*-value obtained from the pathway enrichment analysis (**C**). Metabolic pathway analyses by MetaboAnalyst 5.0 software depict the impact of H_2_O_2_ treatment, with the *x*-axis, *y*-axis, color, or size of the circle reflecting the information as described in Figure 2C.

**Figure 4 antioxidants-12-00452-f004:**
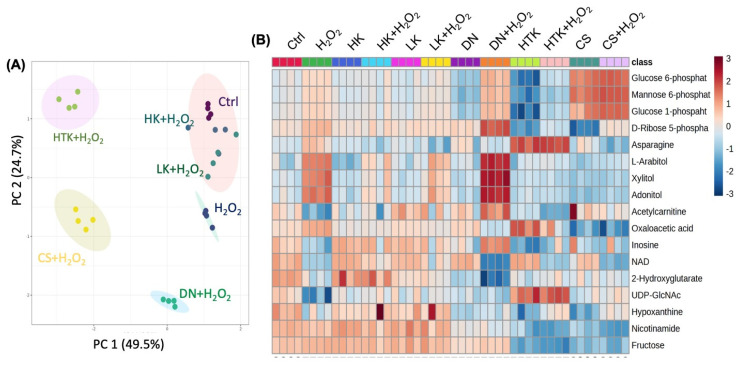
Effect of Cardioplegic Solutions on Metabolites Changed by H_2_O_2_ Treatment. AC 16 cells at 90% confluency were serum-starved in 0.5% FBS DMEM/F12 overnight. The cells were treated with HK, LK, DN, HTK, or CS for 3 h as described in the Section 2, followed by exposure to 200 μM of H_2_O_2_ for 1 h before extraction of aqueous metabolites by 80% methanol for LC-MS/MS-based metabolomics. K-means clustering shows the overall metabolic profile altered by H_2_O_2_ and pretreatment with each cardioplegia solution (**A**, n = 4). The heatmap displays the metabolites in the 9 metabolic pathways significantly altered by H_2_O_2_ treatment for all groups of comparisons (**B**, n = 4).

**Figure 5 antioxidants-12-00452-f005:**
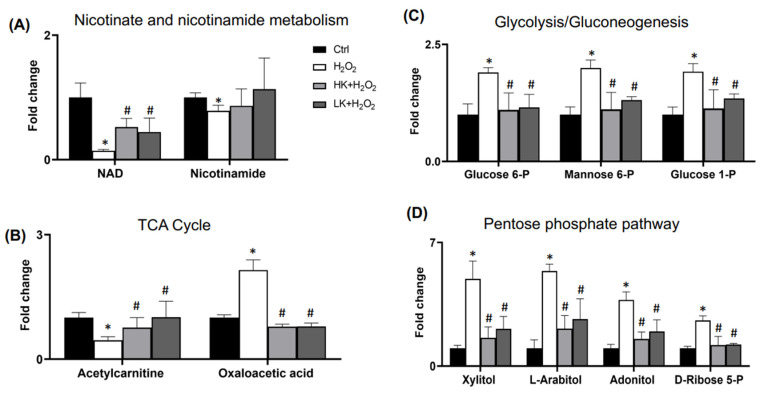
HK or LK Protects against H_2_O_2_ from Inducing Changes in the Metabolites. The metabolites altered by H_2_O_2_ treatment belonging to nicotinate and nicotinamide metabolism, TCA cycle, glycolysis/gluconeogenesis, and pentose phosphate pathway were analyzed for the protective effect of HK or LK cardioplegia solution. * indicates significant difference (n = 4, *p* < 0.05) between control and H_2_O_2_-treated cells, whereas # indicates significant difference (n = 4, *p* < 0.05) between HK or LK with H_2_O_2_ treatment in comparison with H_2_O_2_ treatment alone. These data were analyzed for the significant differences by one-way ANOVA using GraphPad Prism 9 software.

**Figure 6 antioxidants-12-00452-f006:**
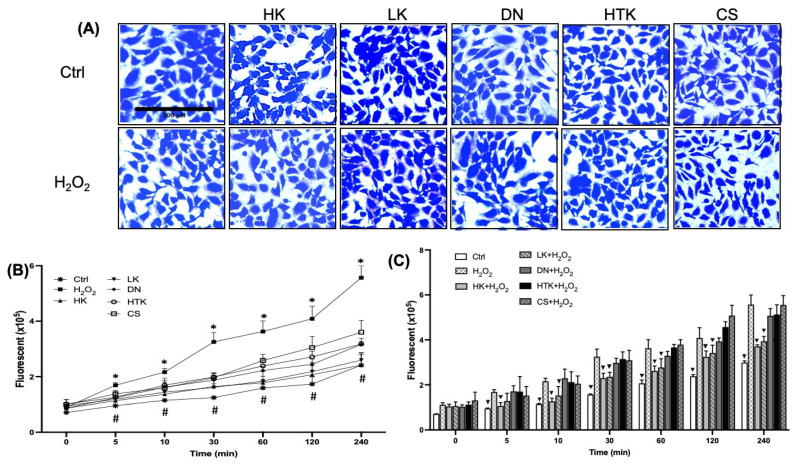
Effects of Cardioplegic Solutions on Cell Morphology and ROS Generation. AC16 cells were seeded in 6-well (**A**) or 96-well plates (**B**,**C**) for pretreatment with cardioplegic solutions for 3 h. After 1 h exposure to 200 μM H_2_O_2_, cell morphology was recorded under an inverted light microscope after fixing and staining in 0.1% Coomassie blue (**A**). Following pretreatment with a cardioplegic solution, the cells were incubated with 100 μM DCFH-DA for 30 min before recording ROS generation with 200 μM H_2_O_2_ treatment by a fluorescence microplate reader (**B**,**C**). The bar graphs represent the means ± SD from 4 replicates of one representative experiment (**C**). * indicates a significant difference (*p* < 0.05) of H_2_O_2_-treated cells from Ctrl or a cardioplegic-solution-treated cells, whereas # indicates a significant difference (*p* < 0.05) of Ctrl from DN-, HTK-, or CS-treated cells (**B**). 
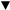
 indicates significant difference from H_2_O_2_-treated cells (**C**). These data were analyzed for the significant differences by one-way ANOVA using GraphPad Prism 9 software.

**Figure 7 antioxidants-12-00452-f007:**
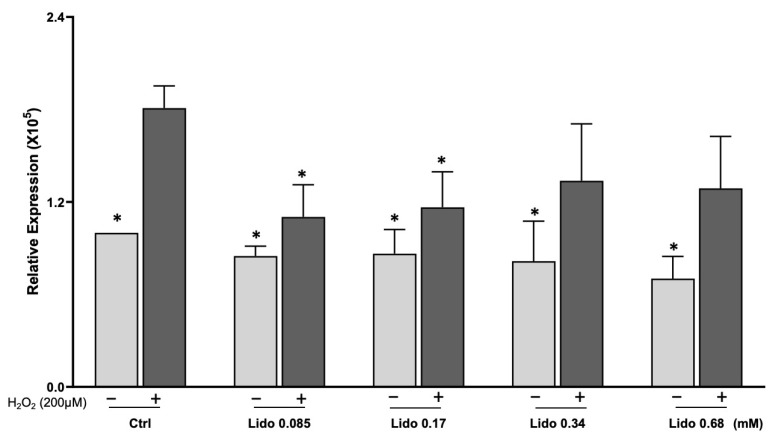
Lidocaine prevents ROS generation by H_2_O_2_. AC16 human cardiac myocytes were seeded in 96-well plates and were pretreated with indicated concentrations of lidocaine for 3 h before 30 min incubation with 10 μM DCFH-DA. H_2_O_2_ was added to the final 200 μM for recording the fluorescence of hydroxy radical formation. The bar graphs represent the means ± SD from 3 independent experiments. * indicates statistical significance (*p* < 0.05) from H_2_O_2_ -treated alone group as determined by two-way ANOVA.

**Table 1 antioxidants-12-00452-t001:** Key Components of Common Cardioplegic Solutions.

	Ratio (Blood: Crystalloid)		Con (mM)	K^+^	Ca^2+^	Na^+^	Mg^2+^	Mannitol	Lidocaine	Histidine
		Type								
**Blood** **cardioplegia**	4:1	**HK**		19.3	1.3	147	7.0	–	0.085	–
	4:1	**LK**		9.3	0.8	153	1.2	–	0.085	–
	1:4	**DN**		25.6	0.2	152.6	15.8	3.6	0.444	–
**Crystalloid cardioplegia**	–	*** HTK**		10	0.02	15	4	–	–	198
	–	**** CS**		15	0.25	100	13	60	–	30

* HTK contains 2 mM tryptophan and 1 mM ketoglutarate; ** CS contains 20 mM glutamate and 3 mM glutathione. The concentrations were calculated on the basis of the dilution of HK, LK, and DN in DMEM/F12.

## Data Availability

The authors confirm that the data supporting the findings of this study are available within the article and its Appendix A. Raw data that support the findings of this study are available from the corresponding author, upon reasonable request.

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
