# Peer review of "Hyperkalemic or Low Potassium Cardioplegia Protects against Reduction of Energy Metabolism by Oxidative Stress"

_antioxidants, 2023, doi:10.3390/antiox12020452_

Round 1
Reviewer 1 Report
Major concerns:
1. Table 1 is missing from the manuscript. This table is critical for information of components of all five solutions. It must be provided.
2. Title: Current title "Hyperkalemic cardioplegia ......…" needs to be changed. It does not reflect the conclusion of this study. From the results, both HK and LK preserve the cardiac metabolism under basic condition or oxidative stress.
3. Results: Figure 1A heatmap needs to be replaced with a high-definition image. The names of molecules were not easily recognized.
4. In addition to metabolic data, morphology and function results of cardiac myocytes merit the quality of this study.
Minor concerns:
None
Author Response
We would like to thank the reviewers for the enthusiasm for this manuscript. We have revised the manuscript to incorporate both reviewers’ comments.
Open Review
(x) I would not like to sign my review report
( ) I would like to sign my review report
English language and style
( ) English very difficult to understand/incomprehensible
( ) Extensive editing of English language and style required
( ) Moderate English changes required
(x) English language and style are fine/minor spell check required
( ) I don't feel qualified to judge about the English language and style
Yes |
Can be improved |
Must be improved |
Not applicable |
|
Does the introduction provide sufficient background and include all relevant references? |
(x) |
( ) |
( ) |
( ) |
Are all the cited references relevant to the research? |
(x) |
( ) |
( ) |
( ) |
Is the research design appropriate? |
(x) |
( ) |
( ) |
( ) |
Are the methods adequately described? |
(x) |
( ) |
( ) |
( ) |
Are the results clearly presented? |
(x) |
( ) |
( ) |
( ) |
Are the conclusions supported by the results? |
(x) |
( ) |
( ) |
( ) |
Comments and Suggestions for Authors
Response: English language and style have been edited by a native English speaking editor.
Major concerns:
- Table 1 is missing from the manuscript. This table is critical for information of components of all five solutions. It must be provided.
Response: Our apology. Table 1 is now included in the submission.
- Title: Current title "Hyperkalemic cardioplegia ......…" needs to be changed. It does not reflect the conclusion of this study. From the results, both HK and LK preserve the cardiac metabolism under basic condition or oxidative stress.
Response: Title has been revised to incorporate this comment.
- Results: Figure 1A heatmap needs to be replaced with a high-definition image. The names of molecules were not easily recognized.
Response: The names of the metabolites in Figure 1A heatmap have been typed in for clarity, since the metabolomics software was not able to produce high-definition images for metabolite labeling.
- In addition to metabolic data, morphology and function results of cardiac myocytes merit the quality of this study.
Response: The morphology data and data on protection against ROS generation have been added during revision (Fig 6).
Reviewer 2 Report
This a well written and intesting publication of significant clinical impact.
Major points
1) Table 1 is missing in the pdf document for review.
2) It is not clear to the reviewer why the cells had to be serum-starved before the experiment. This treatment may cause metabolic changes in the cells.
3) The number of experiments in each of the figures is not mentioned. This should be done to estimate the quality of the statistical analysis.
4) The authors discuss that the protective effect of HK and LK media on H2O2-mediated oxidative stress may be due to the presence of lidocaine in the media. To clarify this point the authors should perform experiments in the absence of lidocaine.
5) Would lidocaine change the outcome of the experiments with HTK and CS?
Author Response
We would like to thank the reviewers for the enthusiasm for this manuscript. We have revised the manuscript to incorporate both reviewers’ comments.
Open Review
(x) I would not like to sign my review report
( ) I would like to sign my review report
English language and style
( ) English very difficult to understand/incomprehensible
( ) Extensive editing of English language and style required
( ) Moderate English changes required
(x) English language and style are fine/minor spell check required
( ) I don't feel qualified to judge about the English language and style
Yes |
Can be improved |
Must be improved |
Not applicable |
|
Does the introduction provide sufficient background and include all relevant references? |
(x) |
( ) |
( ) |
( ) |
Are all the cited references relevant to the research? |
(x) |
( ) |
( ) |
( ) |
Is the research design appropriate? |
(x) |
( ) |
( ) |
( ) |
Are the methods adequately described? |
(x) |
( ) |
( ) |
( ) |
Are the results clearly presented? |
(x) |
( ) |
( ) |
( ) |
Are the conclusions supported by the results? |
(x) |
( ) |
( ) |
( ) |
Response:
English language and style have been edited by a native English speaking editor.
Comments and Suggestions for Authors
This a well written and intesting publication of significant clinical impact.
Major points
- Table 1 is missing in the pdf document for review.
Response: Our apology. Table 1 is now included in the submission.
- It is not clear to the reviewer why the cells had to be serum-starved before the experiment. This treatment may cause metabolic changes in the cells.
Response: The cardioplegic solutions tested in this study are used for adult cardiothoracic surgeries. Since cardiac cells are post-mitotic soon after birth, we used serum starvation to produce post-mitotic state of AC16 cardiomyocytes, in an effort to mimic post-mitotic cardiomyocytes in surgical patients.
- The number of experiments in each of the figures is not mentioned. This should be done to estimate the quality of the statistical analysis.
Response: There are 4 biological replicates for each tested cardioplegic solution for the metabolomics, as shown in the graph. The number of biological replicates has now been added to the figure legend.
- The authors discuss that the protective effect of HK and LK media on H2O2-mediated oxidative stress may be due to the presence of lidocaine in the media. To clarify this point the authors should perform experiments in the absence of lidocaine.
Response: The cardioplegic solutions were obtained as the left-over from cardiothoracic surgery. Because we are not able to obtain all components for cardioplegic solutions as a basic science laboratory, we are not able to make our own lidocaine depleted version.
- Would lidocaine change the outcome of the experiments with HTK and CS?
Response: This is a good point. We have tested whether lidocaine is able to protect against ROS generation and found it indeed does. It will take months to redo metabolomic experiments with lidocaine supplemented HTK or CS. We hope that the lidocaine experiment performed here will address the point.